

# Neighborhood-based inference and restricted Boltzmann machine for microbe and drug associations prediction

Xiaolong Cheng[1], Jia Qu[1], Shuangbao Song[1] and Zekang Bian[2]

[1] School of Computer Science and Artificial Intelligence, Changzhou University, Changzhou, Jiangsu, China
[2] School of AI & Computer Science, Jiangnan University, Wuxi, Jiangsu, China

## ABSTRACT

**Background**. Efficient identification of microbe-drug associations is critical for drug development and solving problem of antimicrobial resistance. Traditional wet-lab method requires a lot of money and labor in identifying potential microbe-drug associations. With development of machine learning and publication of large amounts of biological data, computational methods become feasible.

**Methods**. In this article, we proposed a computational model of neighborhood-based inference (NI) and restricted Boltzmann machine (RBM) to predict potential microbe-drug association (NIRBMMDA) by using integrated microbe similarity, integrated drug similarity and known microbe-drug associations. First, NI was used to obtain a score matrix of potential microbe-drug associations by using different thresholds to find similar neighbors for drug or microbe. Second, RBM was employed to obtain another score matrix of potential microbe-drug associations based on contrastive divergence algorithm and sigmoid function. Because generalization ability of individual method is poor, we used an ensemble learning to integrate two score matrices for predicting potential microbe-drug associations more accurately. In particular, NI can fully utilize similar (neighbor) information of drug or microbe and RBM can learn potential probability distribution hid in known microbe-drug associations. Moreover, ensemble learning was used to integrate individual predictor for obtaining a stronger predictor.

**Results**. In global leave-one-out cross validation (LOOCV), NIRBMMDA gained the area under the receiver operating characteristics curve (AUC) of 0.8666, 0.9413 and 0.9557 for datasets of DrugVirus, MDAD and aBiofilm, respectively. In local LOOCV, AUCs of 0.8512, 0.9204 and 0.9414 were obtained for NIRBMMDA based on datasets of DrugVirus, MDAD and aBiofilm, respectively. For five-fold cross validation, NIRBMMDA acquired AUC and standard deviation of $0.8569 \pm -0.0027$, $0.9248 \pm -0.0014$ and $0.9369 \pm -0.0020$ on the basis of datasets of DrugVirus, MDAD and aBiofilm, respectively. Moreover, case study for severe acute respiratory syndrome coronavirus 2 (SARS-CoV-2) showed that 13 out of the top 20 predicted drugs were verified by searching literature. The other two case studies indicated that 17 and 17 out of the top 20 predicted microbes for the drug of ciprofloxacin and minocycline were confirmed by identifying published literature, respectively.

Corresponding author
Jia Qu, tb17060015b4@cumt.edu.cn

# INTRODUCTION

Studies revealed that microbe communities, primarily include bacteria, viruses, fungi, archea and protozoa, which are closely related to host in human body (*Sommer & Bäckhed, 2013*). Microbe communities inhabit human body organs such as mouth, skin and gastrointestinal tract (*Ventura et al., 2009*). Usually, microbes are considered as a "forgotten organ" for human due to microbes can produce important vitamins, prevent pathogenic invasion, promote metabolic capability and improve immunity (*Gill et al., 2006*; *Kau et al., 2011*; *O'Hara & Shanahan, 2006*; *Smith, McCoy & Macpherson, 2007*). Recently, a growing number of biological and clinical studies have indicated that the imbalance of microbe communities can cause diverse noninfectious diseases (*Bao, Jiang & Huang, 2017*; *Khan et al., 2016*). For example, imbalance of gut microbiota can cause colorectal cancer (*Gagnière et al., 2016*). Decrease of microbe *Bacteroidetes* and increase of microbe *Firmicutes* can lead to obesity (*Ley et al., 2005*). Therefore, it is no surprise that maintaining the balance of microbial communities is essential for human health (*ElRakaiby et al., 2014*).

Moreover, increasing evidence demonstrates that microbes are emerging as novel potential biomarkers or diagnostic/therapeutic tools for disease, laying groundwork for antiviral drug development (*Brown & Hazen, 2017*; *Cummings & Relman, 2000*; *Fazius, Zaehle & Brock, 2013*). In nowadays, drug development faces three main challenges. First, the development cycle of new drugs is long. Usually, a new drug needs an time of 10–15 years from the start to derive marketing approval (*Berdigaliyev & Aljofan, 2020*). Second, the pharmaceutical industry faces multiple problems including high costs of research and development, high failure rates and low productivity (*Khanna, 2012*). Third, a burgeoning number of cases demonstrated that antimicrobial drug resistance has emerged posing significant trouble for drug development and treatment of disease (*Ramirez et al., 2016*). For example, from 1980–2000, the prevalence of drug-resistant *Streptococcus pneumoniae* increased 60-fold with 51% of them resistant to penicillin and 8% of them resistant to third-generation cephalosporin (*Bain & Wittbrodt, 2001*). Thus, it was difficult to treat pneumococcal pneumonia with penicillin and third-generation cephalosporin (*Ament, Jamshed & Horne, 2002*). In Europe, 6% of *K. pneumoniae* were resistant to carbapenems in bloodstream infections which has a high mortality rate of 40–70% (*Ben-David et al., 2012*; *Schwartz & Morris, 2018*). The emergence of antimicrobial drug resistance causes a great threat to humans. Around the world, antimicrobial resistance already caused 700,000 deaths per year and antimicrobial resistance will lead to 10 million deaths per year after 2050 according to the study (*Tagliabue & Rappuoli, 2018*). To solve these problems, drug combination therapies have been employed for fighting antimicrobial drug resistance (*Fischbach, 2011*). Besides, drug repositioning is also an effective method for fighting antimicrobial drug resistance, which can use existing drugs to treat new diseases (*Jarada,*

*Rokne & Alhajj, 2020*; *Xue et al., 2018*). It is worth mentioning that the known microbe-drug association information is crucial for implementation of drug combination and drug repositioning. Therefore, it is an urgent need to develop effective methods to identify potential microbe-drug associations.

Based on the development of sequencing technologies and data acquisition tools, a large number of biological databases have been established over the past few decades, such as GenBank, the Kyoto Encyclopedia of Genes and Genomes (KEGG) and the DNA Data Bank of Japan (DDBJ) (*Chen et al., 2020*; *Kumar & Shanker, 2018*; *Mahmud et al., 2021*). Meanwhile, machine learning has become one of the most rapidly growing technical fields and can be used for a large number of data processing tasks with low-cost computing (*Carleo et al., 2019*; *Jordan & Mitchell, 2015*). Therefore, with the explosion of biological data and low-cost computing driven by machine learning, computational approaches have been widely applied in the diagnosis and treatment of diseases such as horrible cancer (*Cheng et al., 2019*). For example, *Stark et al. (2019)* developed six different machine learning-based models to implement five-year breast cancer risk prediction based on highly accessible personal health data. Those models include logistic regression, Gaussian naive Bayes, decision tree, linear discriminant analysis, support vector machine and feed-forward artificial neural network. *Auffenberg et al. (2019)* developed a random forest machine learning model to provide a prediction of prostate cancer treatment decisions for new patients by using clinical registry data. In particular, the predictive models mentioned above can be integrated into web-based platforms, which brings great convenience to researchers and reduces the cost of medical tests (*Sumathy et al., 2010*).

Because traditional wet-lab method is time-consuming and costly in identifying new microbe-drug associations. Some computational models based on deep learning have been constructed for identifying potential microbe-drug associations. For example, *Long et al. (2020a)* presented a computational model of Graph Convolutional Network (GCN) to predict potential Microbe-Drug Associations (GCNMDA). First, they constructed a heterogeneous network by integrating drug similarity network, microbe similarity network and microbe-drug association network. Then, the random walk with restart was used for microbe similarity network and drug similarity network to obtain a new feature matrix. Subsequently, they used GCN to learn embeddings of nodes based on heterogeneous network and feature matrix. Moreover, they employed conditional random field (CRF) in the hidden layer of GCN for enhancing the node representation learning. They also added attention mechanism into the CRF to accurately aggregate representations of neighborhoods. *Long et al. (2020b)* also developed model of Ensemble framework of graph attention networks (GAT) for microbe-drug association prediction (EGATMDA). First, they constructed three microbe-drug networks (graphs) including microbe-drug bipartite network, microbe-drug heterogeneous network and microbe-disease-drug heterogeneous network based on multiple biological data including drug-drug associations, microbe-microbe associations, known microbe-drug associations, drug-disease associations, microbe-disease associations and disease-disease associations. Second, they constructed a feature matrix by using microbe sequence similarity and drug Gaussian kernel similarity and drug structure similarity. Third, by using graph convolutional network (GCN) and

GAT, nodes embedding representations were learned from feature matrix and each input microbe-drug network. Finally, they removed irrelevant noise *via* using graph-level attention and aggregated the learned node embedding representations to reconstructed a microbe-drug matrix for predicting potential microbe-drug associations. Moreover, *Deng et al. (2021)* proposed a computational model of variational graph autoencoder (VGAE) and deep neural network (DNN) to predict potential Microbe-Drug Association (Graph2MDA). First, they build multi-modal attributed graphs based on drug structure similarity, drug Gaussian kernel similarity, microbe functional similarity, microbe sequence attribute (similarity). Then, they took multi-modal attribute graphs as input and employed VGAE to learn the latent representations of nodes. Finally, they used deep neural network classifier to predict potential microbe-drug associations based on learned embedding obtained by VAGE.

In addition, several computational models based on machine learning were developed to predict potential drugs for SARS-CoV-2 through virus-drug association prediction. For example, *Wang et al. (2021)* developed a model of Gaussian kernel similarity and bounded nuclear norm regularization (BNNR) to predict potential virus-drug association (VDA-GBNNR). First, they build a heterogeneous network based on virus similarity network, drug similarity network and known virus-drug association network. Second, they defined an adjacency matrix to represent constructed heterogeneous network. Then, BNNR, a matrix completion method, was employed to identify new microbe-drug associations by minimizing nuclear norm of adjacency matrix. Recently, *Meng et al. (2021)* proposed a model of similarity constrained probabilistic matrix factorization (called SCPMF) to identify potential virus-drug associations. First, they projected known virus-drug associations matrix into virus feature matrix and drug feature matrix. Second, they introduced drug similarity and virus similarity as constraints for drug feature matrix and virus feature matrix, respectively. Third, gradient descent algorithm was used to obtain final drug feature matrix and virus feature matrix through an iterative process. Finally, the potential virus-drug association matrix was obtained by multiplying transposition of drug feature matrix and virus feature matrix.

Moreover, some network-based computational models were constructed for predicting potential microbe-drug associations. For example, *Peng et al. (2021)* developed a model of Random Walk with Restart (RWR) to predict new virus-drug association (VDARWR). First, they constructed a heterogeneous network by employing virus similarity network, drug similarity network and known virus-drug association. Subsequently, based on heterogeneous network, RWR was used to compute the potential association probabilities between viruses and drugs by using restart probability and computed transition probability of random walk. *Zhou et al. (2020)* developed a model of Virus-Drug Associations by using KATZ to predict drugs against SARS-CoV-2 (VDAKATZ). They first constructed virus-drug heterogeneous networks based on virus similarity, drug similarity and known virus-drug associations. Then, based on the constructed network, a length-based algorithm of KATZ was used to predict potential virus-drug associations by the integration of all walks of different lengths between virus and drugs. Finally, remdesivir, oseltamivir and zanamivir were predicted as the top three potential drugs for SARS-Cov-2 through implementing

VDAKATZ. *Long & Luo (2021)* developed a model of heterogeneous network embedding representation framework for microbe-drug association prediction (HNERMDA). In the model, they constructed a heterogeneous network based on many biological data including microbe-microbe associations, drug-drug associations and known microbe-drug associations. Based on the heterogeneous network, they employed metapath2vec to learn embedding representations for microbes and drugs to more efficiently save microbe-drug association information. In particular, they added a bias network projection recommendation algorithm to identifying new microbe-drug associations more accurately through distributing different bias weights between microbes and drugs.

In this article, we developed a new computational model of neighborhood-based inference (NI) and restricted Boltzmann machine (RBM) for predicting potential microbe-drug association (NIRBMMDA) based on known microbe-drug associations, integrated drug similarity and integrated microbe similarity. First, NI was used to obtain two potential microbe-drug associations matrices by computing associations of similar drugs of drugs with microbes and associations of similar microbes of microbes with drugs, respectively. Then, new microbe-drug associations were predicted by integrating two potential microbe-drug associations matrices. Second, RBM was used to predict potential microbe-drug associations *via* efficiently extracting hidden information from known microbe-drug associations. To improve generalization ability of model, ensemble learning was employed to integrate NI and RBM for predict final potential microbe-drug associations. Moreover, we implemented global leave-one-out cross validation (LOOCV), local LOOCV and five-fold cross validation to evaluate the ability of NIRBMMDA based on the three datasets including DrugVirus, MDAD and aBiofilm, respectively. As a result, the area under the receiver operating characteristics curves (AUCs) of global LOOCV are 0.8666, 0.9413 and 0.9557 for three datasets, respectively. The AUCs of local LOOCV are 0.8512, 0.9204, and 0.9414 for three datasets, respectively. For five-fold cross validation, the average AUCs and the standard deviations are $0.8569 \pm 0.0027$, $0.9248 \pm 0.0014$ and $0.9369 \pm 0.0020$ for three datasets, respectively. Furthermore, three case studies were performed to evaluate the performance of NIRBMMDA. The result showed that 13 out of the top 20 predicted drugs for SARS-CoV-2 were confirmed by searching literature. The other two case studies showed that 17 and 17 out of the top 20 predicted microbes for ciprofloxacin and minocycline were verified by finding published literature, respectively.

## MATERIALS & METHODS

### Microbe-drug association

In this article, three different datasets of MDAD (*Sun et al., 2018*), aBiofilm (*Rajput et al., 2018*) and DrugVirus (*Andersen et al., 2020*) were used to test predictive ability of NIRBMMDA. The MDAD dataset presented in the model contains 2,470 associations between 1,373 drugs and 173 microbes that was collected from MDAD database (*Sun et al., 2018*). Furthermore, the aBiofilm dataset used in the model includes 2,884 associations between 1,720 drugs and 140 microbes collected from aBiofilm database (*Rajput et al., 2018*). Also, *Andersen et al. (2020)* built the DrugVirus database for exploration and
**Table 1  The statistics of three microbe-drug associations datasets.**

| Datasets | Microbes | Drugs | Associations |
|----------|----------|-------|--------------|
| MDAD | 173 | 1373 | 2470 |
| aBiofilm | 140 | 1720 | 2884 |
| DrugVirus | 95 | 175 | 933 |

analysis of broad-spectrum antiviral drugs, in which summarized experimentally verified virus-drug associations. Therefore, the dataset of DrugVirus built here includes 933 associations between 175 drugs and 95 viruses. The statistics of three datasets above are shown in Table 1. Here, we built an adjacency matrix $A \in R^{nd \times nm}$ to preserve microbe-drug association information, where $nd$ represents the number of drugs and $nm$ denotes the number of microbes. If drug $d_i$ associated with microbe $m_j$, the value of entity $A_{ij}$ is 1. Otherwise, the value is 0.

$$A_{ij} = \begin{cases} 1, & \text{if durg } d_i \text{ associated with microbe } m_j \\ 0, & \text{otherwise} \end{cases}. \tag{1}$$

## Drug structural similarity

In this article, SIMCOMP2 search was employed to compute the drug structural similarity (*Hattori et al., 2010*). SIMCOMP2 search (https://www.genome.jp/tools/simcomp2/), a chemical structure search server, can provide links to the KEGG PATHWAY database that contains manually drawn pathway maps with information about molecular interaction, reaction and relation (*Wrzodek, Dräger & Zell, 2011*). In SIMCOMP2 search, by mapping drugs of datasets to those in KEGG, we can obtain drug structural similarity with 0.01 of cut off score that filtrate drug structural similarity score of 0.01 or higher (*Long et al., 2020a*). Then, we defined a matrix $DS1$ to save drug structural similarity where element $DS1(i,j)$ denotes the similarity value between drug $d_i$ and drug $d_j$.

## Drug side effect similarity

The drug-side effect association dataset used in this article were downloaded from SIDER (*Kuhn et al., 2016*). SIDER (http://sideeffects.embl.de/), a side effect resource database, collects information on marketed drugs and their recorded adverse drug reactions. In the dataset, we used $N(i)$ to represent the side effect set associated with drug $d_i$ and employed $N(j)$ to indicate the side effect set of drug $d_j$. Based on the assumption that the more side effect two drugs share, the more similar between the two drugs. If two drugs do not have the same side effects, the score of side effect similarity between the two drugs is equal to 0. We applied Jaccard score to compute drug side effect similarity, which described as Eq. (2) (*Gottlieb et al., 2011*). After that, the matrix $DS2$ was defined to save the drugs side effect similarity and the entity $DS2(i,j)$ denotes the side effect similarity between drug $d_i$ and drug $d_j$.

$$DS2 = \text{Jaccard score} = \left| \frac{N_i \cap N_j}{N_i \cup N_j} \right|. \tag{2}$$

## Microbe sequence similarity

In the model, three different datasets for known microbe-drug associations were used. For 95 viruses in the DrugVirus dataset, we downloaded their complete genome sequences from the National Center for Biotechnology Information (NCBI, https://www.ncbi.nlm.nih.gov/) based on FASTA format. Then, we employed MAFFT, a multiple sequence alignment program, to align the genome sequence of viruses (*Katoh et al., 2002*). Since the MAFFT introduces the approximate distance calculation algorithm and the fast Fourier alignment algorithm, its performs well in accuracy of alignments compared with other multiple sequence alignment software including TCoffee version 2 and CLUSTAL W (*Katoh et al., 2005*). Based on aligned genome sequence of virus, we further used BioEdit to derive the virus sequence similarity matrix. BioEdit, a gratis sequence analysis tool, can compute sequence similarity matrix by using the function of sequence identify matrix (*Tippmann, 2004*). Specially, for microbes in MDAD dataset or aBiofilm dataset, because of the lack of complete genome sequences in NCBI for nearly all microbes, we downloaded another FASTA format of whole genome shotgun sequence of microbe in NCBI. Then, microbe sequence similarity can further be calculated based on MAFFT and BioEdit. According to the idea that more common sequence two microbes share, the more similar between the two microbes. Therefore, the value of microbe sequence similarity score is equal to 0 when the two microbes have no common sequence. Here, the matrix $MS$ was defined as microbe sequence similarity matrix and $MS(m_i, m_j)$ represented the sequence similarity value between microbe $m_i$ and microbe $m_j$.

## Gaussian interaction profile kernel similarity for drugs and microbes

According the former study (*Van Laarhoven, Nabuurs & Marchiori, 2011*), we computed Gaussian interaction profile kernel similarity for drugs and microbes based on the known microbe-drug association matrix $A$. First, we used $IV(d_i)$ to denotes the $i-th$ row vector of matrix $A$ and $IV(m_j)$ to represent the $j-th$ column vector of matrix $A$. Then, Gaussian interaction profile kernel similarity for drugs and microbes can be computed by using Eqs. (3) and (4), respectively. Here, $\|IV(d_i) - IV(d_j)\|$ can be considered as the Euclidean distance for $IV(d_i)$ and $IV(d_j)$. Similarly, $\|IV(m_i) - IV(m_j)\|$ can represent Euclidean distance for $IV(m_i)$ and $IV(m_j)$.

$$GD(d_i, d_j) = \exp\left(-\beta_d \|IV(d_i) - IV(d_j)\|^2\right) \tag{3}$$

$$GM(m_i, m_j) = \exp\left(-\beta_m \|IV(m_i) - IV(m_j)\|^2\right) \tag{4}$$

where $\|\cdot\|^2$ is 2-norm, the $\beta_d$ and $\beta_m$ are normalized kernel bandwidth and are defined as follows:

$$\beta_d = \beta'_d / \left(\frac{1}{nd}\sum_{i=1}^{nd} \|IV(d_i)\|^2\right) \tag{5}$$

$$\beta_m = \beta'_m / \left(\frac{1}{nm}\sum_{i=1}^{nm} \|IV(m_i)\|^2\right) \tag{6}$$

where $\|\cdot\|^2$ is 2-norm, $\beta_d'$ and $\beta_m'$ are the original bandwidths, them are set as 1.

## Integrated similarity for drugs and microbes

To derive the integrated drug similarity, we combined the drug side effect similarity, drug structural similarity and Gaussian interaction profile kernel similarity of drug. If drug $d_i$ and drug $d_j$ have drug structural similarity or drug side effect similarity, the integrated drug similarity is equals to the average of drug structural similarity and drug side effect similarity. Otherwise, the integrated drug similarity is Gaussian interaction profile kernel similarity of drug. Here, we created matrix $SD$ to save integrated drug similarity. The equation of integrated drug similarity is as follows:

$$SD(d_i, d_j) = \begin{cases} \frac{DS1(d_i,d_j)+DS2(d_i,d_j)}{2} & d_i \text{ and } d_j \text{ have structural similarity or side effect similarity} \\ GD(d_i,d_j) & \text{otherwise} \end{cases}. \quad (7)$$

For integrated microbe similarity, we could obtain the integrated microbe similarity by integrating the microbe sequence similarity and Gaussian interaction profile kernel similarity of microbe. The integrated formula is as follows:

$$SM(m_i, m_j) = \begin{cases} MS(m_i, m_j) & m_i \text{ and } m_j \text{ have sequence similarity} \\ GM(m_i, m_j) & \text{otherwise} \end{cases}. \quad (8)$$

## NIRBMMDA

In this article, we proposed a computational model of NIRBMMDA by employing multiple biological data including drug similarity, microbe similarity and known microbe-drug associations. In the model, we carried out NI and RBM to identify potential microbe-drug associations, respectively. Then, an ensemble learning was implemented to integrate the two models for gaining final score of potential microbe-drug associations. The whole flowchart of the NIRBMMDA is shown as Fig. 1. The details of the NIRBMMDA are shown as follows.

## Neighborhood-based inference

The neighborhood-based method is a collaborative filtering algorithm and can recommend potential preference for a user based on preference of similar users (*Su & Khoshgoftaar, 2009*). In this article, we presented a based model of NI for predicting new microbe-drug associations. First, we constructed NI model based on integrated drug similarity. For a drug $d_{i,i=1,2...,nd}$, its neighbors can be obtained by filtering similarity scores based on threshold $\sigma$. The set of neighbors for drug $d_i$ can be defined as $\{d_i | IDS_{u,i} > \sigma, u \neq i\}$. Based on the set above, the potential association score $score1_{i,j}$ between drug $d_i$ and microbe $m_j$ can be obtained by computing the sum associations between the microbe $m_j$ and neighbors of drug $d_i$, which can be described as Eq. (9).

$$score1_{i,j} = \frac{\sum_{i=1}^{nd} \sum_{j=1,u\neq i,IDS_{u,i}\geq\sigma}^{nm} A_{u,j} \times IDS_{u,i}}{\sum_{i=1}^{nd} \sum_{j=1,u\neq i,IDS_{u,i}\geq\sigma}^{nm} IDS_{u,i}} \quad (9)$$

where $nd$ represents the number of drug, $nm$ represents the number of microbe, $A_{u,j}$ denotes the element of the $u-th$ row and $j-th$ column in $A$, and $A^T$ represents the transpose of $A$. The $IDS_{u,j}$ denotes integrated drug similarity between drug $d_u$ and drug $d_j$.
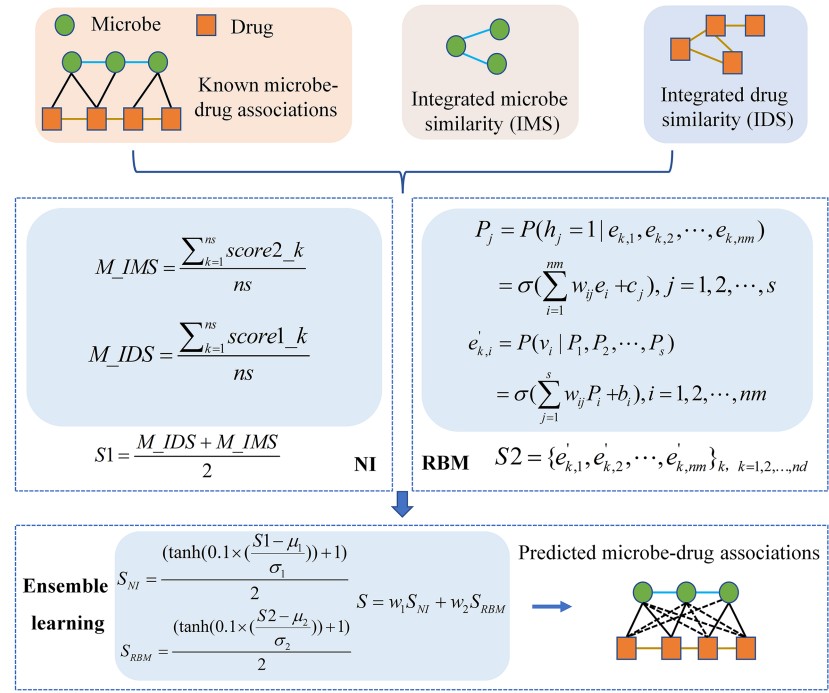

**Figure 1** Flowchart of the computational model of NIRBMMDA.

Since a model of neighborhood-based inference was build based on threshold $\sigma$, we generate multiple thresholds $\{\sigma_1, \sigma_2, \ldots, \sigma_{ns}\}$ to build multiple basic models for reducing the bias of neighbor selection. The value of thresholds $\sigma_i (i = 1, 2, \ldots, ns)$ is between 0 and 0.5 with step size 0.05. After that, an upper bound parameter $\sigma_{upper}$ is used for determining the multiple thresholds that are defined as $\sigma_{threshold} = \{\sigma_i | \sigma_i \leq \sigma_{upper}, i = 1, 2, \ldots, n\} (ns = |\sigma_{threshold}|)$. In this way, $ns$ thresholds $\{\sigma_1, \sigma_2, \ldots, \sigma_{ns}\}$ are used to build $ns$ basic models.

Then, we integrated $ns$ basic models to predict potential microbe-drug associations score by using average strategy, which can be described as follows:

$$M\_IDS = \frac{\sum_{k=1}^{ns} score1\_k}{ns} \tag{10}$$

where $ns$ denotes the number of basic models, $score1\_k$ represents predicted microbe-drug associations score based on the $k-th$ threshold, and $M\_IDS$ denotes potential microbe-drug associations score based on drug similarity.

Moreover, we constructed a NI model based on integrated microbe similarity. The process of building NI model based on integrated microbe similarity is similar to the process of NI model based on integrated drug similarity. For microbe $m_j$, its neighbors can be filtrated through using integrated microbe similarity with threshold $\sigma$. The set of neighbors for microbe $m_j$ is defined as $\{m_j | IMS_{t,j} > \sigma, t \neq j\}$. Based on the set above, the potential association score $score2_{i,j}$ between drug $d_i$ and microbe $m_j$ was obtained by calculating the sum of associations between the drug $d_i$ and neighbors of microbe $m_j$ as

follows:

$$score2_{i,j} = \frac{\sum_{i=1}^{nd} \sum_{j=1,t\neq j,IMS_{t,j}\geq\sigma}^{nm} A_{i,t} \times IMS_{t,j}}{\sum_{i=1}^{nd} \sum_{j=1,t\neq j,IMS_{t,j}\geq\sigma}^{nm} IMS_{t,j}} \qquad (11)$$

where $nm$ represents number of microbe, $nd$ represents number of drug and $A_{i,t}$ denotes the element of the $i-th$ row and $t-th$ column of $A$. The $IMS_{t,j}$ denotes integrated microbe similarity between microbe $m_t$ and microbe $m_j$.

We used multiple thresholds to build multiple basic models for reducing the bias caused by neighbor selection. Then, we integrated multiple basic models to predict potential microbe-drug associations score by using average strategy, which can be described as follows:

$$M\_IMS = \frac{\sum_{k=1}^{ns} score2\_k}{ns} \qquad (12)$$

where $ns$ denotes number of basic models and $score2\_k$ represents predicted microbe-drug associations score based on the $k-th$ threshold and $M\_IMS$ denotes potential microbe-drug associations score based on microbe similarity.

At last, we obtained final prediction score $S1$ based on integrated drug similarity and integrated microbe similarity, which can be described as follows:

$$S1 = \frac{M\_IDS + M\_IMS}{2}. \qquad (13)$$

## Restricted Boltzmann machine model

Restricted Boltzmann Machine (RBM), a stochastic neural network, can be used to learn potential probability distribution (*Smolensky, 1986*). Recently, RBM have been used in numerous fields including movie recommendation, image identification, speech recognition and association prediction in bioinformatics (*Hinton, 2012*; *Wang & Zeng, 2013*). In this article, we employed RBM to build a based model for predicting potential microbe-drug associations. As depicted in Fig. 2, RBM is a two-layer network including visible layer and hidden layer, where each layer includes many units. For a RBM, assume that there is a total of $nm$ visible layer units and $s$ hidden layer units. We used $\mathbf{v} = (v_i, v_2, \ldots, v_{nm})$ to denote set of visible layer units and employed $\mathbf{h} = (h_1, h_2, \ldots, h_s)$ to denote set of hidden layer units. Because there is no intra-layer connection for visible layer units or hidden layer units of the RBM, the energy function between $\mathbf{v}$ and $\mathbf{h}$ can be defined as follows.

$$E(\mathbf{v},\mathbf{h}) = -\sum_{i=1}^{nm} b_i v_i - \sum_{j=1}^{s} c_j h_j - \sum_{i=1}^{i=nm} \sum_{j=1}^{s} w_{ij} v_i h_j \qquad (14)$$

where $nm$ denotes number of visible layer units, $s$ represents number of visible layer units, $b_i$ is bias of $i-th$ visible layer unit $v_i$, $c_j$ is bias of $j-th$ hidden layer unit $h_j$ and $w_{ij}$ represents weight between $v_i$ and $h_j$.

Based on Eq. (14), we obtained marginal distribution over visible layer units by following equation.

$$P(\mathbf{v}) = \sum_{\mathbf{h}} P(\mathbf{v},\mathbf{h}) = \frac{1}{Z} \sum_{\mathbf{h}} e^{-E(\mathbf{v},\mathbf{h})} \qquad (15)$$

 

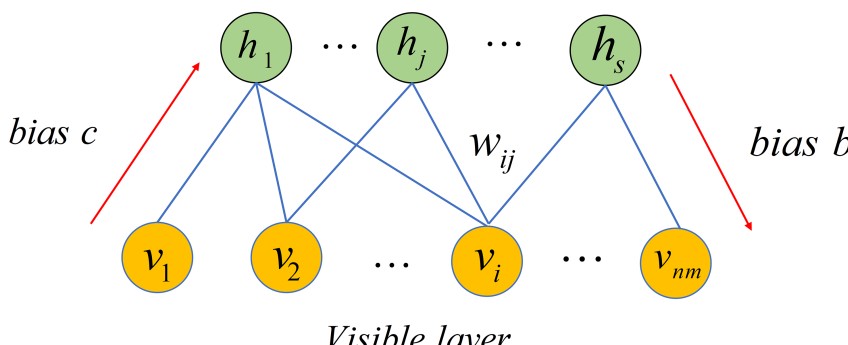

*Hidden layer*

bias c    $w_{ij}$    bias b

*Visible layer*

**Figure 2  Structure diagram of a restricted Boltzmann machine.**

where $Z$ is called the partition function as follows.

$$Z = \sum_{i=1}^{nm} \sum_{j=1}^{s} e^{-E(v_i, h_j)}. \tag{16}$$

Because distributions of units in each layer of RBM are independent, the conditional probabilities of visible layer units and hidden layer units can be defined respectively as follows.

$$P(v_i = 1 | \mathbf{h}) = \sigma \left( \sum_{j=1}^{s} w_{ij} h_j + b_i \right) \tag{17}$$

$$P(h_j = 1 | \mathbf{v}) = \sigma \left( \sum_{i=1}^{nm} w_{ij} v_i + c_j \right) \tag{18}$$

where $\sigma(x) = 1/(1 + e^{-x})$ is sigmoid function.

Given a dataset with $nm$ microbes, $nd$ drugs and known microbe-drug associations, a RBM with $nm$ visible layer units and $s$ hidden layer units is built to predict potential microbe-drug associations. For each drug, the observation $d_k = \{e_{k,1}, e_{k,2}, \ldots, e_{k,nm}\}$ with binary value denotes whether drug $d_k$ is associated with $nm$ microbes. For example, value of $e_{k,1}$ is 1 when drug $d_k$ is associated with microbe $m_1$. Finally, $nd$ drugs have $nd$ observations. When predicting potential associated microbes for a drug, its observation is employed as input of RBM. After that, the prediction is conducted by following two equations.

$$P_j = P(h_j = 1 | e_{k,1}, e_{k,2}, \ldots, e_{k,nm}) = \sigma \left( \sum_{i=1}^{nm} w_{ij} e_{k,i} + c_j \right), j = 1, 2, \ldots, s \tag{19}$$

$$e'_{k,i} = P(v_i | P_1, P_2, \ldots, P_s) = \sigma \left( \sum_{j=1}^{s} w_{ij} P_i + b_i \right), i = 1, 2, \ldots, nm \tag{20}$$

where $\sigma(x) = 1/(1 + e^{-x})$ is sigmoid function. The output of RBM is defined as $score_{k, k=1,2,\ldots,nd} = \{e_{k,1}', e_{k,2}', \ldots, e_{k,nm}'\}$ which denotes the predicted association score between drug $d_k$ and $nm$ microbes. Here, we defined $S2$ to save the predicted microbe-drug associations score.

## Ensemble learning

Because generalization ability of individual predictor is poor, ensemble learning usually is used to integrate the several wake predictors to obtain a more stronger predictor (*Zhou, 2009*). Over the last decades, ensemble learning has been successfully employed to solve many problems including feature selection, computer-aided medical diagnosis and text categorization (*Keyvanpour & Imani, 2013*; *Mohebian et al., 2017*; *Polikar, 2012*). In this article, we used ensemble learning to integrate NI and RBM for inferring potential microbe-drug associations. To obtain common scale score ranged from 0 to 1, prediction scores of NI and RBM are normalized by following two functions (*Polikar, 2006*).

$$S_{NI} = \frac{(\tanh(0.1 \times (\frac{S1-\mu_1}{\sigma_1}))+1)}{2} \tag{21}$$

$$S_{RBM} = \frac{(\tanh(0.1 \times (\frac{S2-\mu_2}{\sigma_2}))+1)}{2} \tag{22}$$

where $\mu_1$ and $\sigma_1$ are mean and standard deviation of scores produced by the NI. Similarly, $\mu_2$ and $\sigma_2$ are mean and standard deviation of scores produced by the RBM. Subsequently, the different weights were allocated for NI and RBM to derive better prediction performance. Here, we created matrix $S$ to save the potential microbe-drug association score as follows.

$$S = w_1 S_{NI} + w_2 S_{RBM} \tag{23}$$

where $w_1$ is weight for NI and $w_2$ is weight for RBM. The sum of $w_1$ and $w_2$ is 1.

## RESULTS

### Performance evaluation

We employed global LOOCV, local LOOCV and five-fold cross validation to evaluate the predicted performance of NIRBMMDA based on the three datasets of DrugVirus (*Andersen et al., 2020*; *Long et al., 2020a*), MDAD (*Sun et al., 2018*) and aBiofilm (*Rajput et al., 2018*), respectively. In LOOCV, each known microbe-drug association was selected in turn as test sample and remaining known microbe-drug associations were used as training samples. For global LOOCV, all unknown microbe-drug pairs were employed as candidate samples. Then, we used training samples to train model and used the trained model to predict score of test samples and candidate samples. We further ranked test sample with candidate samples based on predicted scores in global LOOCV. At last, we obtained the ranking of all test samples. In local LOOCV, score of test sample was ranked with scores of candidate samples which included the investigated drug of the test samples. At last, we also obtained the ranking of all test samples. In five-fold cross validation, the known microbe-drug associations were randomly divided into five subsets where each subset was regarded as test sample in turn and other four subsets were considered as training samples. All unknown microbe-drug pairs would be treated as candidate samples. Subsequently, we ranked score of each test sample with scores of candidate samples. Finally, we obtained the ranking of

**Table 2   AUC and standard deviation (SD) of ensemble learning (EL) in 11 groups of weights for NI and RBM based on dataset of DrugVirus, MDAD and aBiofilm.** Bolded values indicate the best result in 11 groups of results.

| Datasets | EL | The | 11 | | Groups | Weights | | | | | | |
|---|---|---|---|---|---|---|---|---|---|---|---|---|
| | group**s** | 1 | 2 | 3 | 4 | 5 | 6 | 7 | 8 | 9 | 10 | 11 |
| | NI | 1 | 0.9 | 0.8 | 0.7 | 0.6 | 0.5 | 0.4 | 0.3 | 0.2 | 0.1 | 0 |
| | RBM | 0 | 0.1 | 0.2 | 0.3 | 0.4 | 0.5 | 0.6 | 0.7 | 0.8 | 0.9 | 1 |
| DrugVirus | AUC | 0.8282 | 0.8464 | 0.8540 | 0.8568 | **0.8569** | 0.8552 | 0.8521 | 0.8521 | 0.8478 | 0.8424 | 0.8261 |
| | SD | 0.0040 | 0.0032 | 0.0029 | 0.0027 | **0.0027** | 0.0026 | 0.0027 | 0.0027 | 0.0027 | 0.0028 | 0.0035 |
| MDAD | AUC | 0.9169 | 0.9246 | **0.9248** | 0.9239 | 0.9226 | 0.9209 | 0.9190 | 0.9167 | 0.9139 | 0.9099 | 0.9021 |
| | SD | 0.0018 | 0.0015 | **0.0014** | 0.0014 | 0.0013 | 0.0013 | 0.0012 | 0.0012 | 0.0011 | 0.0011 | 0.0012 |
| aBiofilm | AUC | 0.9323 | 0.9364 | **0.9369** | 0.9363 | 0.9354 | 0.9343 | 0.9329 | 0.9313 | 0.9291 | 0.9259 | 0.9183 |
| | SD | 0.0028 | 0.0024 | **0.0020** | 0.0018 | 0.0016 | 0.0015 | 0.0015 | 0.0015 | 0.0014 | 0.0014 | 0.0014 |

all test samples. Particularly, the five-fold cross validation was performed 100 times for avoiding bias caused by random sample divisions. If the ranking of test sample exceeded the given threshold, NIRBMMDA would be considered to achieve a correct prediction. Further, receiver operating characteristics (ROC) curve was plotted through true positive rate (TPR, sensitivity) against false positive rate (FPR, 1-specificity) at diverse thresholds. Sensitivity is the proportion of the test samples which rank over the pre-set threshold, while the specificity is the percentage of candidate samples whose ranking are lower than the appointed threshold. AUC could be used to evaluate the predictive performance of NIRBMMDA. The NIRBMMDA's prediction is random when the value of AUC is 0.5. If the AUC's value is 1, the predictive result of NIRBMMDA is perfect.

Because NIRBMMDA integrated two based model of NI and RBM, weights of NI and RBM would affect the performance of NIRBMMDA. We tested 11 group of wights of NI and RBM with a range from 0 to 1 (step size 0.1) for three datasets of DrugVirus, MDAD and aBiofilm respectively based on five-fold cross validation (see Table 2). For DrugVirus, the result showed that NIRBMMDA obtained the best performance of AUC and standard deviation with 0.8569 ±0.0027 when weight of NI is 0.6 and weight of RBM is 0.4. Based on the selected two weights, we compared the performance of NIRBMMDA with other four classical models of HGIMDA (*Chen et al., 2016*), IMCMDA (*Chen et al., 2018a*), KATZMDA (*Chen et al., 2017*) and MDGHIMDA (*Chen et al., 2018b*) according to five-fold cross validation. The evaluation result showed that our model is better than HGIMDA (0.6995 ±0.0024), IMCMDA (0.6776 ±0.0034), KATZMDA (0.8229 ±0.0022) and MDGHIMDA (0.8293 ±0.0033). Then, according to the two selected weights mentioned above, we compared NIRBMMDA with the four identical comparison models based on global LOOCV and local LOOCV, respectively. In the global LOOCV, NIRBMMDA obtained better performance with AUC of 0.8666 than HGIMDA (0.7048), IMCMDA (0.6901), KATZMDA (0.8305), MDGHIMDA (0.8518) (see Fig. 3). In the local LOOCV, the AUC of NIRBMMDA is 0.8512, which is better than HGIMDA (0.7537), IMCMDA (0.7425), KATZMDA (0.8216), MDGHIMDA (0.8509) (see Fig. 3).

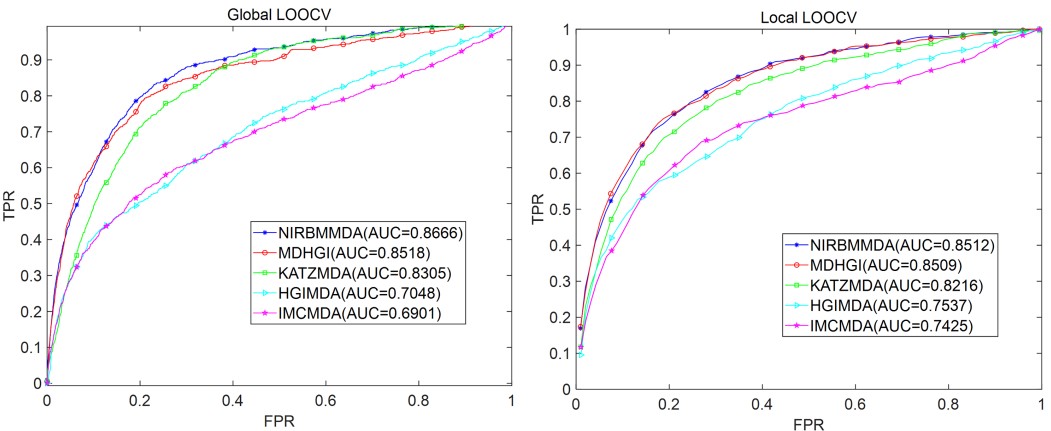

**Figure 3** Comparison of prediction performance between NIRBMMDA and other four models (HGIMDA, IMCMDA, KATZMDA, MDGHIMDA) based on the DrugVirus dataset. (A) In terms of ROC curves and AUCs based on global LOOCV. (B) In terms of ROC curves and AUCs based on local LOOCV.

For MDAD dataset, based on five-fold cross validation, NIRBMMDA obtained the best AUC and standard deviation of 0.9248 ±0.0014 when weight of NI is 0.8 and weight of RBM is 0.2 (see Table 2). As comparison algorithms, AUCs and the standard deviation of HGIMDA (0.8152 ±0.0012), IMCMDA (0.7849 ±0.0025), KATZMDA (0.9173 ±9.6340e−04) and MDGHIMDA (0.8153 ±0.0019) are less than the evaluation result of NIRBMMDA. Then, based on weights used in five-fold cross validation, we calculated AUCs of global LOOCV and local LOOCV for NIRBMMDA, HCIMDA, IMCMDA, KATZMDA and MDGHIMDA, respectively. As a result, in the global LOOCV, NIRBMMDA obtained the AUC with 0.9413, which is better than HCIMDA (0.8173), IMCMDA (0.7891), KATZMDA (0.9247) and MDGHIMDA (0.8446) (see Fig. 4). In the local LOOCV, the AUCs of HCIMDA (0.8301), IMCMDA (0.8035), KATZMDA (0.9119) and MDGHIMDA (0.8537) are less than NIRBMMDA (0.9204) (see Fig. 4).

For aBiofilm dataset, based on five-fold cross validation, the NIRBMMDA obtained the best AUC and the standard deviation with 0.9369 ±0.0020 when NI's weight is 0.8 and RBM's weight is 0.2 (see Table 2). As comparison algorithms, HGIMDA (0.8412 ±0.0014), IMCMDA (0.7509 ±0.0073), KATZMDA (0.9305 ±8.0311e−04), MDGHIMDA (0.8201 ±0.0022) are less than NIRBMMDA (0.9369 ±0.0020). Subsequently, based on the selected two weights mentioned above, we computed the AUCs of global LOOCV and local LOOCV for NIRBMMDA, HCIMDA, IMCMDA, KATZMDA and MDGHIMDA, respectively. In the global LOOCV, NIRBMMDA derived an AUC of 0.9557, which is higher than HCIMDA (0.8482), IMCMDA (0.7584), KATZMDA (0.9378) and MDGHIMDA (0.8491) (see Fig. 5). In the local LOOCV, NIRBMMDA obtained better AUC with 0.9414 than AUCs of other four classical models for HCIMDA, IMCMDA, KATZMDA and MDGHIMDA with 0.8837, 0.7718, 0.9302 and 0.8707 respectively (see Fig. 5).

In summary, NIRBMMDA obtained better prediction accuracy compared with four state-of-the-art models for datasets of DrugVirus, MDAD and aBiofilm based on LOOCV
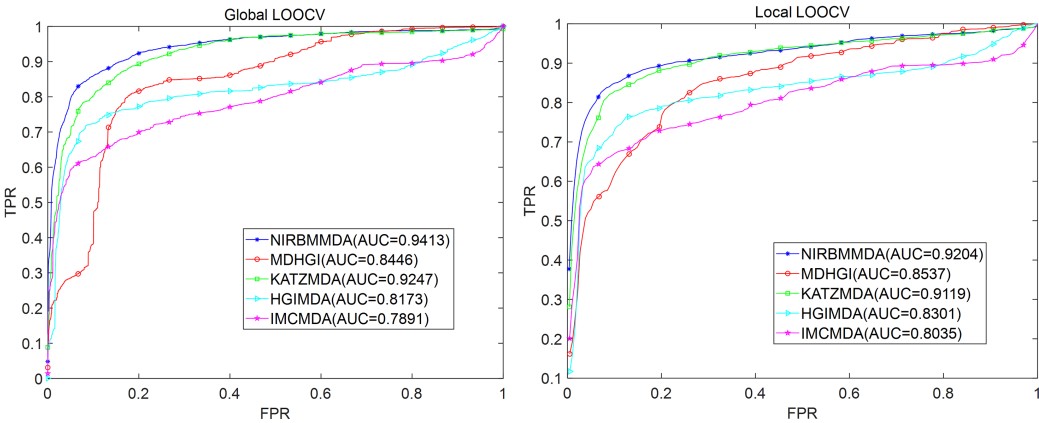

**Figure 4 Comparison of prediction performance between NIRBMMDA and other four models (HGIMDA, IMCMDA, KATZMDA, MDGHIMDA) based on MDAD dataset.** (A) In terms of ROC curves and AUCs based on global LOOCV. (B) In terms of ROC curves and AUCs based on local LOOCV.

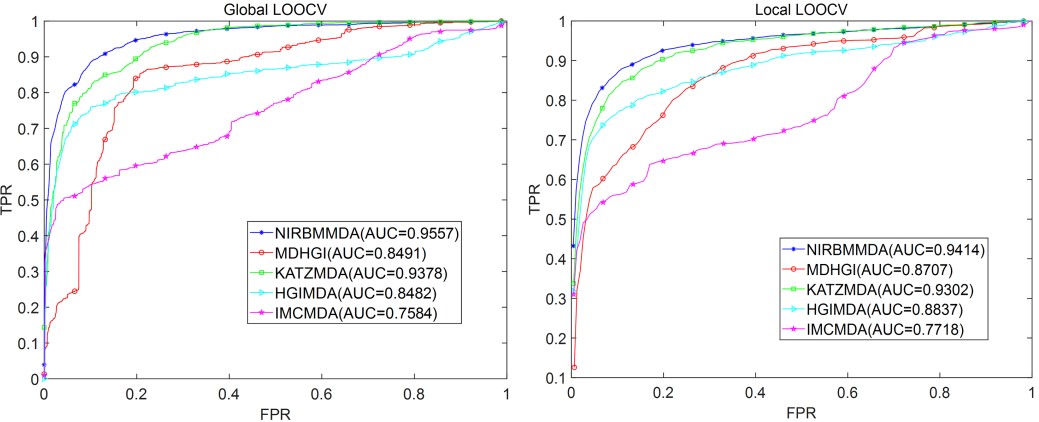

**Figure 5 Comparison of prediction performance between NIRBMMDA and other four models (HGIMDA, IMCMDA, KATZMDA, MDGHIMDA) based on the aBiofilm dataset.** (A) In terms of ROC curves and AUCs based on global LOOCV. (B) In terms of ROC curves and AUCs based on local LOOCV.

and five-fold cross validation. These results indicated that NIRBMMDA has an outstanding and stable performance in predicting potential microbe-drug associations.

## Discussing parameters of model

For NIRBMMDA, there are some key parameters needed to be determined including threshold $\sigma_{upper}$ used in NI. According to the previous study (*Zhang et al., 2016*), we tested 10 candidate values of $\sigma_{upper}$ with a range from 0.05 to 0.5 (step 0.05) and calculated corresponding 10 AUPR scores based on five-fold cross validation on the training samples. After that, the $\sigma_{upper}$ with the best AUPR score was selected to predict potential microbe-drug associations based on test sample.

**Table 3  Computational procedures of the contrastive divergence (CD) algorithm.**

| Algorithm: CD |
| --- |
| **Input:** training set of $N$ batch $v_{n=1}^N$, number of hidden units $s$, number of visible units $nm$ |
| **Output:** $b_i, c_j, w_{ij}$ |
| **1 Initialize:** $b_i = 0, c_j = 0, w_{ij}$ was randomly initialized, $k = 50, \varepsilon = 0.1$ |
| **2 for** $t = 1,2,\ldots,k$ **do** |
| **3**   **for** each batch $v_n$, n=1,2,\ldots,N **do** |
| **4**       **for** $j = 1,2,\ldots,s$ **do**   // from visible layer to hidden layer |
| **5**       $P(h_j^* = 1|v) = \sigma(w_{ij}v_i + c_j)$ |
| **6**       **end for** |
| **7**       **for** $i = 1,2,\ldots,nm$ **do**      // from hidden layer to visible layer |
| **8**       $P(v_i^* = 1|h) = \sigma(w_{ij}h_j + b_i)$ |
| **9**       **end for** |
| **10**       $w_{ij} = w_{ij} + \varepsilon * [p(h_i = 1|v)v_j^T - p(h_i^* = 1|v^*)v_j^{*T}]$ |
| **11**       $b_j = b_j + \varepsilon(v_j - v_j^*)$ |
| **12**       $c_i = c_i + \varepsilon * [p(h_i = 1|v) - p(h_i^* = 1|v^*)]$ |
| **13**   **end for** |
| **14 end for** |

Similarly, for another parameter $s$ used in RBM, we tested 11 candidate values ranging from 20 to 120 with step size 10, computed corresponding 11 AUPR scores and employed the $s$ with the best AUPR score to carry out prediction. Moreover, visible layer units bias $b_i$, hidden layer units bias $c_j$ as well as weight $w_{ij}$ between $i-th$ visible layer unit $v_i$ and $j-th$ hidden layer unit $h_j$ are also used in RBM. The contrastive divergence (CD) algorithm (*Hinton, 2002*) is employed to determined $b_i$, $c_j$ and $w_{ij}$ on the basis of training temples. The specific process of the CD algorithm is illustrated in Table 3.

## Case studies

To further validate the prediction performance of NIRBMMDA, we implemented three type of case studies. In first type of case study, we predicted potential drugs for SARS-COV-2 through implementing NIRBMMDA based on the DrugVirus dataset. For the second and third type of case studies, we predicted potential microbes for ciprofloxacin and minocycline through implementing NIRBMMDA based on the MDAD dataset and the aBiofilm dataset, respectively. Our main research interest is in computational bioinformatics. Therefore, we usually confirmed the predicted results presented in case study by databases and published literatures. For some predicted association information that is not validated by any study, we hope the predicted associations can be further confirmed by biologist based on biological experiments in the future.

SARS-COV-2 is a kind of coronavirus with high transmission efficiency, which emerged at the end of 2019 and posed a huge threat to human health (*Hu et al., 2021*; *Hui et al., 2020*; *Wu, Leung & Leung, 2020*). SARS-CoV-2 can cause severe respiratory lesions and lung damages after entering cells (*Zhu et al., 2020*). Therefore, it is an urgent need to found effective drugs for SARS-CoV-2. *Hu, Frieman & Wolfram (2020)* found that chloroquine may have effect for treating for COVID-19 caused by SARS-CoV-2 based on study of nanomedicine. Moreover, *Shannon et al. (2020)* found that favipiravir can exerts an antiviral effect for SARS-CoV-2 by slowing RNA synthesis. Here, we used
**Table 4  Prediction of the top 20 predicted drugs associated with SARS-COV-2 based on the DrugVirus dataset.** The first column records top 1–10 related drugs. The second column records the top 11–20 related drugs.

| Drug name | Evidence | Drug name | Evidence |
|---|---|---|---|
| Erlotinib | Unconfirmed | Inosine pranobex | PMID: 33339426 |
| Didanosine | Unconfirmed | Cidofovir | PMID: 33594342 |
| Amiodarone | PMID: 32737841 | Alisporivir | PMID: 32409832 |
| Idoxuridine | PMID: 34188314 | Aciclovir | Unconfirmed |
| Azacitidine | Unconfirmed | Anisomycin | PMID: 33289002 |
| Glycyrrhizin | PMID: 33918301 | Amantadine | PMID: 33040252 |
| Berberine | PMID: 33670363 | Irbesartan | PMID:33735271 |
| Amprenavir | PMID: 34344455 | ABT-263 | Unconfirmed |
| Labyrinthopeptin A1 | Unconfirmed | Foscarnet | Unconfirmed |
| Doxycycline | PMID: 32873175 | Darunavir | PMID: 32889701 |

NIRBMMDA to predict potential drug for SARS-COV-2. Then, we ranked predicted drugs for SARS-COV-2 based on predicted score and further verified the top 20 potential drugs by finding literatures on PubMed. The result showed 13 of the first 20 predicted drugs for SARS-COV-2 were verified (see Table 4). For example, the association between SARS-COV-2 and idoxuridine was predicted and ranked fourth. Idoxuridine is a nucleoside analog and have been used as an antiviral drug for herpes (*Almalki et al., 2021*). *Almalki et al. (2021)* found that idoxuridine has significant antiviral activity for SARS-COV-2 through using molecular docking. Moreover, the association between SARS-COV-2 and glycyrrhizin was predicted and ranked sixth. Glycyrrhizin, also named glycyrrhizic acid, is a bioactive substance extracted from a medicinal herb of glycyrrhiza (*He et al., 2019*). *Yu et al. (2021)* found that glycyrrhizin was an efficient and nontoxic anti-SARS-COV-2 drug by using computer-aided drug design and biological verification.

Ciprofloxacin, second generation fluoroquinolone, shows outstanding antimicrobial activity with few side effects for treating bacterial infections over 30 years (*Zhang et al., 2018*). In this article, we employed NIRBMMDA to predict ciprofloxacin-related microbes. Then, we ranked the ciprofloxacin-related microbes according to predicted score and confirmed the top 20 potential associated microbes for ciprofloxacin by finding the literature on PubMed. The result showed that 17 out of the top 20 ciprofloxacin-related microbes were confirmed (see Table 5). For example, the top-ranked microbe for ciprofloxacin is *Serratia marcescens*. *Serratia marcescens*, a Gram-negative and non-sporulating bacillus, could cause lung infection, otitis and sepsis (*Veraldi & Nazzaro, 2016*). *Veraldi & Nazzaro (2016)* found ciprofloxacin can treat skin ulcers caused by *Serratia marcescens* through investigating three patients in hospital. Moreover, the association between *Mycobacterium avium* and ciprofloxacin was predicted and ranked third. *Mycobacterium avium* is an environmental microbe which exists in water, soil, bird and mammal hosts (*Sangari, Parker & Bermudez, 1999*). *Klopman et al. (1993)* found ciprofloxacin show activity against the *Mycobacterium avium* by using the microdilution method.

**Table 5 Prediction of the top 20 predicted microbes associated with Ciprofloxacin based on the MDAD dataset.** The first column records top 1–10 related microbes. The second column records the top 11–20 related microbes.

| Microbe name | Evidence | Microbe name | Evidence |
|---|---|---|---|
| Serratia marcescens | PMID:27052490 | Klebsiella pneumoniae | PMID: 27257956 |
| Candida albicans | PMID:19109335 | Streptococcuspneumoniae serotype 4 | Unconfirmed |
| Mycobacterium avium | PMID: 8239587 | Vibrio harveyi | PMID: 27247095 |
| Clostridium perfringens | PMID: 24944124 | Enterococcus faecium | PMID: 30015506 |
| Human immunodeficiency virus 1 | PMID: 9566552 | Enterococcus faecalis | PMID: 30015506 |
| Enteric bacteria and other eubacteria | PMID: 31321030 | Staphylococcus epidermidis | PMID: 9111541 |
| Streptococcus | PMID: 30502964 | Plasmodium falciparum | PMID: 31451506 |
| Listeria monocytogenes mutans | PMID: 22003016 | Actinomyces oris | Unconfirmed |
| Streptococcus pneumoniae | PMID: 12917240 | Proteus mirabilis | PMID:26953206 |
| Human immunodeficiency virus | Unconfirmed | Candida spp. | PMID:30781782 |

Minocycline, second generation tetracycline derivative, has good antibacterial activity (*Jonas & Cunha, 1982*; *Nagarakanti & Bishburg, 2016*). In addition, minocycline has been found to have non-antibiotic effects for inflammatory diseases based on open clinical trials (*Garrido-Mesa, Zarzuelo & Gálvez, 2013*). Particularly, minocycline has emerged effect in neuroprotection demonstrated by various studies in animal models (*Garrido-Mesa, Zarzuelo & Gálvez, 2013*; *Romero-Miguel et al., 2021*). Therefore, minocycline has been used for treating acne and could be a potential drug for neurodegenerative and inflammatory diseases such as dermatitis, Parkinson's disease and Alzheimer's disease (*Garrido-Mesa, Zarzuelo & Gálvez, 2013*; *Romero-Miguel et al., 2021*). In this case study, *via* the implementation of NIRBMMDA, we can predict potential microbes associated with drug of minocycline. Subsequently, we sorted predicted microbes for minocycline according to the predicted score and verified the top 20 potential microbes by finding the published literature. The result showed that 17 out of the top 20 microbes for minocycline were confirmed (see Table 6). Among the top 20 predicted microbes for minocycline, *Pseudomonas aeruginosa* was predicted with the first ranking. *Pseudomonas aeruginosa*, a common Gram-negative bacterium, can lead to severe infections for human (*Chevalier et al., 2017*). *Chen et al. (2019)* found that minocycline possessed antimicrobial activity for *Pseudomonas aeruginosa in vitro* experiment. Furthermore, the association between *Streptococcus mutans* and minocycline was predicted and ranked third. *Streptococcus mutans* possesses strong virulence factors including high acid production, ability to form compact biofilm and production of glucans (*Abdel-Aziz, Emam & Raafat, 2020*). *Baker et al. (1983)* found that minocycline can inhibit plaque formation caused by *Streptococcus mutans in vitro* pure cultures.

## DISCUSSION

Because the emergence of antimicrobial drug resistance and long development cycle of new drugs, an increasing number of researchers have been focused on the problem of potential association prediction between microbes and drugs based on computational models. In
**Table 6 Prediction of the top 20 predicted microbes associated with Minocycline based on the aBiofilm dataset.** The first column records top 1–10 related microbes. The second column records the top 11–20 related microbes.

| Microbe name | Evidence | Microbe name | Evidence |
| --- | --- | --- | --- |
| Pseudomonas aeruginosa | PMID: 30817887 | Salmonella enterica | PMID: 34475718 |
| Candida albicans | PMID: 28367877 | Streptococcus pyogenes | PMID: 28161292 |
| Streptococcus mutans | PMID: 6580410 | Vibrio harveyi | PMID: 28252178 |
| Escherichia coli | PMID: 30129883 | Listeria monocytogenes | PMID: 30267005 |
| Staphylococcus epidermis | PMID: 30226742 | Streptococcus sanguis | Unconfirmed |
| Staphylococcus epidermidis | PMID: 8592428 | Actinomyces oris | PMID: 29782813 |
| Enterococcus faecalis | PMID: 32944085 | Corynebacterium ammoniagenes | Unconfirmed |
| Serratia marcescens mutans | PMID: 25468904 | Aggregatibacter actinomycetemcomitans | PMID: 21405933 |
| Bacillus subtilis | PMID: 34124228 | Pseudomonas libaniensis | Unconfirmed |
| Vibrio cholerae | PMID: 28062293 | Burkholderia pseudomallei | PMID: 15509614 |

this article, we proposed a computational model of NIRBMMDA to identify potential microbe-drug associations by using ensemble learning method based on NI and RBM.

The outstanding performance of NIRBMMDA mainly come from the following several key factors. First, NI and RBM was used as based predictors. NI can efficiently utilize similarity data to predict new microbe-drug associations by adopting different thresholds to filtering neighbors. RBM is a two-layer generative stochastic artificial neural network that can effectively extract the latent features of known microbe-drug associations. Second, experimentally confirmed microbe-drug associations used in the model were downloaded from three highly reliable databases including DrugVirus, MDAD and aBiofilm. In addition, some reliable biological data used in the model, including Gaussian interaction profile kernel similarity for drugs and microbes, drug side effect similarity, drug structural similarity and microbe sequence similarity, which can greatly increase predicted accuracy of the model. Third, the success of NIRBMMDA also follows the implementation of ensemble learning which can integrate weak predictor including NI and RBM for obtaining a stronger predictor.

However, there are still some limitations in NIRBMMDA that need to be overcome in the future. First, the number of experimentally confirmed microbe-drug associations from databases of DrugVirus, MDAD and aBiofilm is not enough. More known microbe-drug associations need to be confirmed by experiment, which can further improve predicted accuracy of NIRBMMDA. Second, some microbes lack genome sequences on NCBI, which would influence predicted accuracy of the proposed model. We hope that the missing microbe genome sequences will be experimentally measured in the future. Third, the two based predictor used in the model may not be enough and more based predictor are employed may contribute to improve predicted accuracy.

## CONCLUSIONS

We proposed a model named NIRBMMDA to predict potential microbe-drug association. In the model, NI and RBM were used to predict potential microbe-drug associations, respectively. Considering generalization ability of individual model may be poor, we

used an ensemble learning method to predict potential microbe-drug associations through integrating predicted associations matrices of NI and RBM. Moreover, we used LOOCV and five-fold cross validation to evaluate performance of NIRBMMDA based on three datasets including DrugVirus, MDAD, aBiofilm. Results indicated that NIRBMMDA obtained better performance compared with HCIMDA, IMCMDA, KATZMDA and MDHGI. Further, implementation of three case studies for SARS-COV-2, drug ciprofloxacin and drug minocycline illustrated that NIRBMMDA is an effective prediction model. Although NIRMBMDA achieved good predictive performance in case studies, it may depend on the database or the choice of microbes. As it is known, the dataset used to train the model can affect the performance of the model. The more known microbe-drug associations, the higher the accuracy of the model. Moreover, the number of known microbe-drug associations is different for datasets of DrugVirus, MDAD and aBiofilm, which affects the Gaussian interaction profile kernel similarity inputted into the model and leads to different prediction performance for model on three datasets. Also, abundant similarity data can contribute to improve the prediction accuracy of model.

## ACKNOWLEDGEMENTS

This study is part of Xiao-Long Cheng's Masters in Computer Science work.

### Funding

The authors received the following grants: Scientific Research Foundation of Jiangsu Provincial Education Department: 21KJB520030; Natural Science Foundation of Jiangsu Province grant number: BK20220621; 2020 scientific research start-up fund: ZMF20020461; and Postgraduate Research & Practice Innovation Program of Jiangsu Province:KYCX22_3062. The funders had no role in study design, data collection and analysis, decision to publish, or preparation of the manuscript.

### Grant Disclosures

The following grant information was disclosed by the authors:
Scientific Research Foundation of Jiangsu Provincial Education Department: 21KJB520030; 2020.
Scientific research start-up fund: ZMF20020461.
Postgraduate Research & Practice Innovation Program of Jiangsu Province: KYCX22_3062.
Natural Science Foundation of Jiangsu Province: BK20220621.

### Competing Interests

The authors declare there are no competing interests.

### Author Contributions

- Xiaolong Cheng conceived and designed the experiments, performed the experiments, analyzed the data, prepared figures and/or tables, authored or reviewed drafts of the article, and approved the final draft.

- Jia Qu conceived and designed the experiments, performed the experiments, analyzed the data, prepared figures and/or tables, authored or reviewed drafts of the article, and approved the final draft.
- Shuangbao Song analyzed the data, authored or reviewed drafts of the article, and approved the final draft.
- Zekang Bian analyzed the data, authored or reviewed drafts of the article, and approved the final draft.

## Data Availability

The data and code are available in the Supplemental Files.

## Supplemental Information

Supplemental information for this article can be found online at http://dx.doi.org/10.7717/peerj.13848#supplemental-information.

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
