# Peer review of "Neighborhood-based inference and restricted Boltzmann machine for microbe and drug associations prediction"

_PeerJ, doi:10.7717/peerj.13848_

## Round 0.1 · original submission · Minor Revisions

Dear authors,

After careful analysis by two experts in the field, they concluded that the manuscript presented is well written and contains relevant and interesting information from a theoretical framework. There are minor aspects to be improved. I kindly request that all authors carefully revise the manuscript and address all the comments rise by the reviewers and submit your manuscript again.

I thank you for this very interesting paper and for choosing Peer J.

Best regards,
Bernardo

·

Basic reporting

Studies on the response of pathogens to antibiotic treatment are limited by the time and resources necessary to attend each clinical case individually. A promising strategy is the computerized use of biological information available in databases and the development of bioinformatic tools, which is one of the main objectives of this study.

However, this type of machine learning has already been used for the diagnosis and treatment of diseases such as cancer, the background should include this type of information that allows the reader to have confidence and context in the use of computational approaches to the clinic.

On the other hand, the quality of the figures should be improved, in particular, figure 1 and 2.
The objective of this article is clear and relevant, the strategy used to test the system and validate it in a bibliographical way and with experimental data is acceptable. The results presented are consistent and in accordance with the objective of the investigation.

Experimental design

The topic of this article is clear, well defined and relevant. The application of this research can be significant and enriching in the study of antibiotic resistance and treatment of various pathogens. The methodology used fully meets the requirements for the computational study of interactions and the methodology used and developed is described in sufficient detail to be replicated and used by other researchers. Validation of the proposed strategy with other pathologies generated by non-emerging microorganisms would be useful.

Validity of the findings

The study is of benefit to the area of knowledge, the objectives and conclusions are well expressed. The data used are robust, however they can be enriched and validated with more experimental information.

Additional comments

I believe that this article can be accepted after improving the bibliographic information that supports the importance of predictive studies that use machine learning in the medical and microbiological area. In addition to improving the quality and resolution of the figures and including in the conclusions that the proposed proposal works efficiently in the cases described in the article but that it may depend on the databases or microorganisms analyzed.

·

Basic reporting

It is an interesting manuscript in general well written and understandable, references are adequate.
However, bacterial names are not in italics and sometimes the first letter of the gender was not I capital letter, e.g. pseudomonas should be Pseudomonas.

Experimental design

No experiments were done, it is only a theoretical work, validated with existing information from the literature, it is well structured, the research question is well defined and enough details of the methods are provided.

Validity of the findings

The predictions of the developed model were validated using the literature, looking like a robust approximation.

Interestingly some information obtained and sowed in tables 5 and 6 is not validated by any study, like the possible effect of ciprofloxacin over Streptococcus pneumoniae serotype 4, Actinomyces oris and HIV virus, also the predicted antibacterial effects of minocycline over Salmonella enterica and other 3 bacterial species, it would be ideal that at least some of those non validated effects are tested to further validate the strength of the model predictions.

Additional comments

L 533 “Therefore, minocycline has been used for treating many diseases such as acne, neurodegenerative conditions and dermatitis (Nagarakanti & Bishburg 2016).

Please explain why if minocycline is an antibiotic it is used for neurodegenerative conditions and dermatitis

L 540 “the pseudomonas aeruginosa was” remove “the”, capitalize “P”

Also put scientific names in the reference in italics.

---

## Round 0.2 · accepted · Accept

Dear authors,
I thank you for choosing PeerJ. After careful review by two experts, they concluded and I concur, that your study is now suitable for publication. Congratulations!

Best regards,
Bernardo

·

Basic reporting

The comments have been addressed according to the possibilities of the researchers based on the available literature.

Experimental design

The validation of the computer results with available literature that expresses experimental evidence was attended. Table 6 increased by 2.

Validity of the findings

The proposal presented here is relevant in the medical and informatics areas.

·

Basic reporting

it is now OK

Experimental design

it is now OK

Validity of the findings

it is now OK

Additional comments

I thank the authors for addressing my comments and for the modifications made to their manuscript.